# Chemical Composition Analysis and Antioxidant Activity of *Coffea robusta* Monofloral Honeys from Vietnam

**DOI:** 10.3390/foods11030388

**Published:** 2022-01-29

**Authors:** Nguyen Thi Nu Trinh, Nguyen Ngoc Tuan, Tran Dinh Thang, Ping-Chung Kuo, Nguyen Ba Thanh, Le Nhat Tam, Le Hong Tuoi, Trang H. D. Nguyen, Danh C. Vu, Thi L. Ho, Le Ngoc Anh, Nguyen Thi Thu Thuy

**Affiliations:** 1Institute of Biotechnology and Food Technology, Industrial University of Ho Chi Minh City, Ho Chi Minh 700000, Vietnam; nutrinh28218@gmail.com (N.T.N.T.); nguyenngoctuan@iuh.edu.vn (N.N.T.); thanhngba@iuh.edu.vn (N.B.T.); lenhattam@iuh.edu.vn (L.N.T.); letuoii1808@gmail.com (L.H.T.); nguyenhadieutrang@iuh.edu.vn (T.H.D.N.); 2School of Pharmacy, College of Medicine, National Cheng Kung University, Tainan 701, Taiwan; z10502016@ncku.edu.tw; 3Institute of Applied Technology, Thu Dau Mot University, Thu Dau Mot 820000, Vietnam; danhvc@tdmu.edu.vn; 4College of Agriculture and Applied Biosciences, Can Tho University, Can Tho 94000, Vietnam; hlthi@ctu.edu.vn; 5Department of Food Technology, Ho Chi Minh City University of Technology, Vietnam National University-Ho Chi Minh City, Ho Chi Minh 700000, Vietnam; leanh.ftech@gmail.com; 6R&D Department, Vietnam Dairy Products J.S Company, Ho Chi Minh 700000, Vietnam; nttthuy2@vinamilk.com.vn

**Keywords:** *Coffea robusta*, monofloral honey, phenolic acid, caffeine, antioxidant activity

## Abstract

Monofloral honey samples (*Coffea robusta*) from Vietnam were determined for their chemical compositions. This is the first report on the chemical composition and antioxidant activity of coffee honey from Vietnam. These samples were characterized by their high contents of total and reducing sugars, total phenolic contents, and total flavonoid contents. The contents of seven phenolic acids (PAs) were quantified by high performance liquid chromatography (HPLC) and analyzed with the assistance of principle component analysis (PCA) to differentiate the honey samples into groups. The hydroxymethylfurfural (HMF) (0.048–2.933 mg/kg) and free acid contents (20.326–31.163 meq/kg) of coffee honey were lower in Nepal, which reflected the freshness of the honey when conducting this survey. The coffee honey had total sugar and reducing sugar contents 831.711 g/kg and 697.903 g/kg, respectively. The high level of total phenolic (0.642 mg GAE/g) and flavonoid (0.0341 mg GE/g) contents of coffee honey contributed to their antioxidant activity of this honey sample. Among the coffee honey tested, the IC_50_ of DPPH radical-scavenging activities value was 1.134–17.031 mg/mL, while the IC_50_ of ABTS radical-scavenging activities value was 115.381–213.769 mg/mL. The phenolic acids composition analysis displayed that gallic acid appeared in high concentrations in all studied honey samples, ranging from 0.037–1.015 mg/kg, and ferulic acid content ranged from 0.193 to 0.276 mg/kg. The content of trigonelline and caffeine in coffee honey samples ranged from 0.314–2.399 mg/kg and 8.946–37.977 mg/kg. The data in this article highlight the relevance of coffee honey as a healthy substance.

## 1. Introduction

Honey is a popular food product around the world. The characteristic features of honey are its sweet taste, pleasant aroma, nutritional properties, and antibacterial and antioxidant activity. Its composition contains over 200 different chemical compounds, including sugar, minerals, proteins, amino acids, and vitamins, and other bioactive ingredients, such as phenolics, flavonoids, etc. The varied characteristic features of honey are derived from the types of floral sources of the crops surrounding the beehives. In addition, the physical and chemical properties of honey are also dependent on the climatic conditions and geographical origin [1]. Many current studies also showed that bioactive compounds in honey, which provide benefits such as antioxidant, anti-inflammatory, antibacterial, antiviral, antidiabetic, and anticancer properties [2,3,4,5,6,7].

Recently, researchers have been interested in the findings of the contents of bioactive compounds from coffee honey. Previous studies reported that coffee honey contains a high content of ascorbic acid and a low content of total flavonoids. Moreover, the caffeine content in honey is eight times higher than that in coffee nectar [8]. However, other researchers have reported a high degree of flavonoids and phenolic contents in samples of Thailand coffee honey. This study also evidenced its antibacterial activity against *Staphylococcus aureus*, *Propionibacterium acnes*, and *Corynebacterium* sp. which usually causes skin diseases, and its significant antioxidant and anti-tyrosinase activities [9]. Therefore, this honey could be applied for the treatment of cold, flu, and skin diseases, and as an immune system booster.

In Vietnam, along with the development of coffee industry, beekeeping for honey in Dak Lak is developing quickly towards industrial farming. Currently, the composition of *Coffea robusta* monofloral honey in Vietnam has not been studied, especially the quality of honey raised in different regions in Dak Lak. In continuation of our ongoing research projects on the chemical composition of coffee honey, this study focused on determining the physicochemical characteristics, total polyphenol content, total flavonoid content, phenolic acids, alkaloids, and antioxidant activity in the honey grown in beekeeping facilities in some communes of Dak Lak province in Vietnam. In addition, the physicochemical parameters and the contents of phenolic acids and alkaloids were analyzed with the assistance of principle component analysis (PCA) to differentiate the honey samples into groups. The results of this study will provide a basis for the judgement of the quality of coffee honey from other regions.

## 2. Materials and Methods

### 2.1. Chemicals and Reagents

The phenolic standards (gallic acid, cinnamic acid, chlorogenic acid, caffeic acid, coumaric acid, ferulic acid, 2,4-dihydroxybenzoic acid), quercetin, caffein, trigonelline, 2,2-diphenyl-1-picrylhydrazyl (DPPH), 2,2′-azino-bis (3-ethylbenzothiazoline-6-sulfonic acid) (ABTS) and Folin–Ciocalteu’s reagent were purchased from Sigma–Aldrich (St. Louis, MO, USA). Chemicals used for the physicochemical analysis were purchased from Merck (Darmstadt, Germany). All chemicals used were analytical grade unless specified. All samples were stored in sealed glass jars and at room temperature (20–30 °C) until analysis

### 2.2. Collection of Coffee Honey Samples

Coffee honey samples were collected at the end of the coffee blossom (March 2021) at seven beekeeping establishments in Cu Kuin district (CoffeeHC 1, CoffeeHC 2, CoffeeHC 3), Buon Me Thuot City (CoffeeHB 1, CoffeeHB 2) and Cu M’gar district (CoffeeHM 1, CoffeeHM 2) of Dak Lak province, Vietnam. In addition, there are three commercial coffee honey samples (Commercial T, Commercial K, Commercial H) purchased in a supermarket and three honey samples of other origins (forest honey (FH), Tithonia diversifolia honey (TH) and longan honey (LH)) purchased in a local market, which were used for comparison. All samples were stored in sealed glass jars and at room temperature (20–30 °C) until analysis.

### 2.3. Physicochemical Analysis

Honey samples were analyzed based on the criteria of total sugar determination by phenol-sulfuric acid method and free reducing sugar contents by DNS method according to the previously reported literature [10]. Hydroxymethylfurfural (HMF) and free acid contents were determined using spectrophotometric method [11], mineral content was quantified by calcination method [12].

#### 2.3.1. Total Sugar and Free Reducing Sugar Contents

Total sugar was determined by the phenol-sulfuric method. Honey (0.5 g) was dissolved in 20 mL of water and then made up to 100 mL with distilled water. The solution was filtered through an 11 µm filter paper. The filtrate (1 mL) was titrated to 100 mL with distilled water. The solution (1 mL) was added with phenol 5% (1 mL) and mixed well, then concentrated H_2_SO_4_ (5 mL) was added and mixed well. After leaving the reaction for 10 min at 24 °C, we continued to cool the solution in a thermostatic bath at 25 °C for 10 min and then measured it photometrically at 490 nm. The experiment was repeated 3 times. Glucose was used to construct the calibration curve with a concentration scale of 0; 1; 2.5; 5; 10; 25; 50; 100 ppm. Total sugar in honey was determined through the standard curve of glucose according to the following formula:(1)$$\mathrm{Total}\;\mathrm{sugar}\;\mathrm{content}\;\left(\%\right)=C\times \frac{V_{1}\times V_{2}\times 100}{10^{6}\times V\times W}$$
C: Total sugar concentration from the calibration curve (ppm);*W*: Weight of sample (g);*V*_1_: 1st titration volume (mL);*V*_2_: Second titration volume (mL);10^6^: Convert ppm concentration to g/mL;*V*: Analytical sample volume (mL).

The free reducing sugars present in honey were determined by the DNS method. Honey (0.5 g) was dissolved in 20 mL of water and then made up to 50 mL with distilled water. The solution was filtered through an 11 µm filter paper. The filtrate (1 mL) was titrated to 25 mL with distilled water. The solution (2 mL) had DNS (1 mL) added to it. The reaction solution was kept in a thermostatic bath at 90 °C until a reddish-brown color appeared. Then it was allowed to cool for 10 min. Then, the reaction solution had distilled water (7 mL) added to it. The absorbance was determined with a spectrophotometer at 540 nm. The experiment was repeated 3 times. Glucose was used to construct the calibration curve with a concentration scale of 0; 1; 2.5; 5; 10; 25; 50; 100 ppm. The reducing sugar content of honey was determined through the standard curve of glucose according to the following formula:(2)$$\mathrm{glu}\left(\%\right)=C\times \frac{V_{1}\times V_{2}\times 10\times 100}{10^{6}\times V\times W}$$
C: Concentration from the calibration curve (ppm);*V*_1_: Volumetric first time (mL);*V*_2_: Second titration volume (mL);10^6^: Convert concentration (ppm) to (g/mL);*V*: Analytical sample volume (mL);*W*: Weight of sample (g).

#### 2.3.2. Hydroxymethylfurfural (HMF)

Honey (5 g) was dissolved in 25 mL of distilled water. The absorbance was measured at 284 and 336 nm against a filtered solution treated with NaHSO_3_. The hydroxymethylfurfural (HMF) content was calculated using the following formula:(3)$$\mathrm{HMF}=\frac{\left(A_{284}-A_{336}\right)\times 149.7\times 5\times D}{W}$$
*D*: dilution factor;*W*: sample weight (g).

#### 2.3.3. Free Acid Contents

Honey (10 g) was added to 75 mL of distilled water, and then the solution was stirred with a magnetic stirrer until the solution was completely dissolved. The solution was titrated with 0.1 N NaOH to pH = 8.30, recording the volume of NaOH consumed.
(4)$$\mathrm{Free}\;\mathrm{acid}\;\mathrm{contents}\;\left(\mathrm{mqe}/\mathrm{kg}\right)=\frac{0.1\times V_{NaOH}\times 1000}{W}$$
*V_NaOH_*: volume of NaOH (ml);*W*: sample weight (g).

#### 2.3.4. Mineral Content

Honey sample (1 g) was put in a crucible, heated at 550 °C for 6 h.

### 2.4. Determination of Total Phenolic Content (TPC) and Flavonoid Content (TFC)

The total polyphenol content (TPC) was determined by Folin–Ciocalteu method [13]. The standard gallic acid solutions (0.01, 0.03, 0.05, 0.07 and 0.09 mg/mL) were used to validate the method. The gallic acid solution (0.01, 0.03, 0.05, 0.07 and 0.09 mg/mL) (1 mL) was incubated with the 10% Folin-Ciocalteu’s reagent (5 mL) for 5 min, and then a Na_2_CO_3_ solution (4 mL, 75 g/L) was further added to the solution. The reaction was maintained at room temperature for 30 min. The absorbance was determined with a spectrophotometer at 765 nm, which was plotted against the corresponding standard concentrations. The honey sample (2.5 g) was mixed with 50 mL of distilled water to reach the concentration of 0.05 mg/mL. It was prepared similarly to that of standard gallic acid solution. The TPC was expressed in mg of gallic acid equivalents (mg GAE/g of honey). The experiments were performed in triplicate.
(5)$$A=11.703\times C+0.039\;(R^{2}=0.9963)$$
where: *A*: absorbance; *C*: concentration.

The total flavonoid content (TFC) was determined by the spectroscopic method according to Jia Zhishen et al. [14]. The quercetin solutions (0.025, 0.050, 0.075, 0.100 and 0.150 mg/mL) were used as the standards to construct a calibration curve. The standard solution (0.5 mL) was incubated with 2.5 mL distilled water and 0.15 mL 5% NaNO_2_ for 5 min, and then it reacted with 0.3 mL of 10% AlCl_3_ for another 6 min. Finally, the mixture was added to 1 mL of 1 M NaOH and 0.55 mL of distilled water. The absorbance was determined with a spectrophotometer at 510 nm. The honey solutions (0.05 mg/mL) were prepared similarly to that of standard gallic acid solution. The TFC was expressed in mg of quercetin equivalents (mg QE/g of honey). The experiments were performed in triplicate.
(6)$$A=0.0101\times C+0.0248\;(R^{2}=0.9961)$$
where: *A*: absorbance; *C*: concentration.

### 2.5. Antioxidant Activity

#### 2.5.1. DPPH Free Radical Scavenging Assay

Honey was dissolved in methanol. The range of investigated concentrations was 1–50 mg/mL. The honey extract (1 mL) was mixed with 3 mL of methanolic solutions containing DPPH radicals (0.01 mM). The mixtures were shaken and left for 30 min in the dark at 24 °C. Then the absorbances were measured at 517 nm. The blank was performed with methanol instead of methanolic extract. The radical-scavenging activity (*RSA*) was calculated as the percentage of DPPH discoloration using the equation:(7)$$\%\;RSA=\frac{A_{0}-A_{S}}{A_{0}}\times 100$$
where: *A*_0_ and *A_S_* are the absorbances of the blank and the sample, respectively.

#### 2.5.2. ABTS Free Radical Scavenging Assay

ABTS was dissolved in methanol to a 7 mM concentration. ABTS radical cation (ABTS^•+^) was produced by reacting ABTS stock solution with potassium persulfate (2.45 mM) at a ratio of 1:1. The mixture was left in the dark at 24 °C for 12–16 h before use. The ABTS^•+^ solution was diluted with methanol to obtain an absorbance of 0.70 (±0.02) at 734 nm. Honey was dissolved in methanol to a range of investigated concentrations, which were 100–300 mg/mL. The honey solution (0.5 mL) was mixed with 5 mL of the dilute ABTS^•+^ solution, and the absorbance was measured after 6 min relative to a blank (water) at 734 nm. The radical-scavenging activity (*RSA*) was calculated as the percentage of ABTS^+•^ discoloration using the Equation (7).

Ascorbic acid (AA) was used as a reference. The IC50 was determined as the concentration of the tested honey sample causing 50% reduction of the initial DPPH or ABTS^•+^ concentration, measured from the linear regression concentration curve of the test extract (mg/mL) against the percentage of the radical scavenging inhibition.

### 2.6. Phenolic Analysis

#### 2.6.1. Extraction of Phenolic Compounds

The phenolic compounds were extracted by the method described by Stanek [15] and Oroian [16]. Honey (1.0 g) was dissolved in 5 mL 50% methanol in acidified water (formic acid, pH 2.2), which homogenized in an ultrasonic bath for 30 min. The resulted solutions were filtered by a SPE C18 cartridge, which was previously activated and equilibrated with methanol (3.0 mL) and acidified deionized water (3.0 mL, pH 2.2). The honey solution (5.0 mL) was taken into the treated cartridge and passed at 1.0 mL/min, and the solid-phase extraction (SPE) cartridges were further washed with 6.0 mL acidified deionized water (pH 2.2) to remove the honey matrices. Finally, the polyphenols remaining in the cartridge were eluted with 6.0 mL methanol and adjusted to the volume of 10.0 mL. The methanol fraction was filtered through a 0.45 µm membrane and stored at −20 °C until further analysis by the liquid chromatographic system.

#### 2.6.2. Preparation of Phenolic Acid Standard Solutions

The standards used to quantify phenolic acids include gallic acid, cinnamic acid, chlorogenic acid, caffeic acid, coumaric acid, ferulic acid, and 2,4-dihydroxybenzoic acid. The standard was prepared at the concentration range between 0.1 and 5.0 ppm. The PA standard (1.0 mg) was dissolved with a mixture of methanol:water (50:50, pH = 2.2) with equal volume of formic acid in a 10 mL volumetric flask. Standard mixture (100 ppm) was prepared by mixing seven PAs in 100 mL volumetric flask. The resulting solution was diluted to the mark with a mixture of methanol:water (50:50, pH = 2.2) using an equal volume of formic acid. The working standard solutions were established at the concentrations of 0.1, 0.25, 0.5, 1, 2.5, 5.0 ppm. The standard solutions were filtered over a 0.45 µm disc.

#### 2.6.3. HPLC Analysis

HPLC was performed on a Shimadzu LC-2030C 3D liquid chromatographic system (Shimadzu, Japan), equipped with a Photodiode array (PDA) detector. The analytical column used was a VertiSep™ GES C18 HPLC column (250 mm × 4.6 mm, 5.0 µm). The operating conditions were as follows: column temperature, 30 °C; flow rate, 0.80 mL/min; injection volume, 10 µL; monitoring wavelength range, 190–450 nm. The elution solvent system consisted of methanol (A) and water/formic acid (pH 2.2) (B) with a concentration gradient program of: 25% A for 3 min, 25–40% A for 5 min, hold 40% A for 5 min, 40–60% A until 16 min, 60% A for 5 min, 60–80% A for 3 min, 80% A for 3 min, then lower to 25% A to the 35th min [17].

### 2.7. Trigonelline and Caffein Analysis

#### 2.7.1. Extraction of Alkaloid Compounds

The honey sample (2 g) was dissolved in 10 mL of distilled water and homogenized in an ultrasonic bath at 40 degrees for 30 min. Then, the solution was extracted liquid–liquid with chloroform (25 mL) and shaken for 12 h. The chloroform was recovered and dried by nitrogen blowing. The dry residue was re-dissolved with methanol (1 mL). Then, the methanol fraction was filtered through a 0.45 µm membrane and injected into the chromatographic system [8].

#### 2.7.2. HPLC analysis

Liquid chromatography was performed on a Shimadzu LC-2030C 3D liquid chromatographic system (Shimadzu, Japan), equipped with a Photodiode array (PDA) detector. The analytical column used was a VertiSep™ GES C18 HPLC column (250 mm × 4.6 mm, 5.0 µm). The solvent system was methanol:water (45:55), the flow rate was 0.4 mL/min and the injection volume was 10 µL. Spectral data from all peaks were collected in the range of 200–400 nm and trigonelline and caffeine was detected at 272 nm.

### 2.8. Data Analysis

The results were reported as mean ± standard deviation (SD). One-way Analysis of Variance (ANOVA) with Tukey’s HSD post hoc test was used to evaluate the differences between all the physio-chemical parameters and content of seven phenolic acids of honey samples. Besides this, Principal component analysis (PCA) method was performed with FactoMineR packages (R software version 4.0, http://factominer.free.fr (accessed on 1 September 2021)) to classify honey groups with specific physicochemical parameters of each group. Finally, linear regression analysis method was used to set up the linear equation standard form of total polyphenol and total flavonoid contents.

## 3. Results

### 3.1. Physicochemical Parameters

In all the analyzed honey samples, the total sugar contents ranged from 742.860 to 884.718 g/kg, among which the average total sugar content of the coffee honey samples was 831.711 g/kg (Table 1). The total sugar content of coffee honey was comparable to that of some honeys from Algeria [18], which are higher than those of honey collected in Pakistan [19] and Benin [20]. The sugar content is responsible for properties such as viscosity, hygroscopicity, and energy value of honey. Honey is an important natural sweetener, having a low glycemic index with almost 80% simple sugars from the total chemical composition (35–40% fructose and 30–35% glucose). Sugar from honey has demonstrated effects on both healthy and diabetic subjects [21], and it has also raised the insulin levels of diabetic humans [22]. Therefore, coffee honey with total sugar content will be able to be a good source of sweetening for diabetic patients. 

The reducing sugar contents in the studied honey samples ranged from 625.620 to 757.769 g/kg (Table 1). As compared with previous studies, the average free reducing sugar content of coffee honey (697.903 g/kg) was higher than that in Malaysian honey (655.3 g/kg) [23] or honey in Benin (580–600 g/kg) [20], so it can be concluded briefly that the free reducing sugar content in coffee honey is relatively high. The higher the reducing sugar content in honey, the more beneficial to health the honey is, especially for diabetics because the main reducing sugar in honey is fructose, which has been shown to reduce the index blood sugar [24], regulate hormone production in the pancreas, and promote its hepatic actions for diabetic patients [25,26]. Thus, the coffee honey contained a relatively high content of total sugar and reducing sugars within the limits of acceptable values and also promises to bring many benefits to consumers, especially diabetic patients. Among the samples of coffee honey, there was a significant difference observed for the total and free reducing sugar content. The sugar content and composition of honey highly depended on the regions and the origins of flowers where honey was collected. With a relatively high content of total and free reducing sugars, coffee honey not only provides an abundant source of energy, but this also serves as an outstanding feature that can be used to distinguish it from other honey samples on the market.

The hydroxymethylfurfural (HMF) is one of the important parameters to determine the freshness of honey, as well as the storage duration and conditions, and it tends to increase during processing and/or aging of the product [27,28]. Several factors have been reported to influence the levels of HMF, such as temperature and time of heating, storage conditions, pH, and floral sources, therefore the content of HMF provides an indication of overheating and storage in poor conditions [29]. The results showed that the 13 honey samples surveyed had HMF contents ranging from 0.049 to 2.933 mg/kg (Table 1). The CoffeeHC 3 sample had the highest HMF content (2.933 ± 0.022 mg/kg) but was still lower than the honey in Nepal (49.5 mg/kg) [30] or the honey from *Melipona subnitida* (7.56 mg/kg) and *Apis mellifera* (10.82 mg/kg) [11]. The contents of HMF in the honey samples were relatively low because the samples were analyzed immediately after being brought to the laboratory and the correct transportation and storage processes also maintained the honey as fresh samples.

The lowest free acid content was detected in the Longan Honey sample (15.220 meq/kg) and the highest in the CoffeeHM 1 sample (31.163 meq/kg) (Table 1). In the coffee honey samples, the free acid content had a statistically significant difference among the samples. The lowest free acid content was detected in sample CoffeeHC 1 (20.164 meq/kg) and the highest in the CoffeeHM 1 sample (31.163 meq/kg). Most of the samples have a statistically significant difference; only samples CoffeeHC 2 and CoffeeHB 2 did not show a statistically significant difference. The acid content of honey is influenced by many factors, such as the time and manner of honey being stored, the presence of different organic acids in the flower, geographical origin, harvest season, etc. Therefore, the difference in free acid content in coffee honey samples can be elucidated well. The average free acid content of coffee honey (26.336 meq/kg) was higher than those of commercial coffee honey (21.8 meq/kg) and other flower honeys (19.729 meq/kg). As compared with the mean free acid content in some previous publications, acacia honey was found to have a very low concentration (5.44 meq/kg); however, luso-type honey (31.2 meq/kg) [31] and honey in Algeria (50.4 meq/kg) [32] had relatively high concentrations. The free acid content in coffee honey is in the average range and may be taken to be indicative of honey freshness because the free acid value can indicate the occurrence of fermentation [33]. The free acids in honey contribute to the flavor and are partially responsible for the antimicrobial activity [34].

The mineral contents in the investigated honey samples ranged from 0.026 to 0.138% (Table 1), consistent with the previous report of Vanhanen et al. [35] about the mineral content in honey fluctuating in the range of 0.04 to 0.2%. The highest mineral content, found in CoffeeHC 3 sample (0.13%), was 3.4 times higher than that of the lowest sample of Forest Honey (0.026%). The average mineral content of the coffee honey (0.09%) was higher than the average mineral content of the commercial coffee honey (0.068%) and the mineral content of other flower honey groups (0.071%). Although there was a difference between the groups in mineral contents, the difference between them was not statistically significant. The previous reports on the mineral content in some types of honey are very high (up to 0.4%, honeydew honey), which is 45 times higher than that of coffee honey, and there are also low grades such as Viper’s bugloss honey (0.013%) that is 6.9 times lower. Compared to the range of mineral content of other honey samples reported previously (0.04 to 0.2%), the mineral content in Dak Lak coffee honey was assessed to be in the medium range.

### 3.2. Total Phenolic Contents (TPC) and Total Flavonoid Contents (TFC)

In the coffee honey samples, the polyphenol contents of CoffeeHC 3 (0.704 mg GAE/g), CoffeeHM 1 (0.749 mg GAE/g), and CoffeeHB 1 (0.730 mg GAE/g) samples were similar and different from the other samples, in which CoffeeHC 1 sample (0.519 mg GAE/g) had the lowest polyphenol content (Table 2). It has already been demonstrated that the polyphenol contents varied due to many factors caused by their growing geographical regions including the temperature, rainfall, flora, and soil [36]. The polyphenol content in honey has been reported previously as being as low as Tulang honey in Malaysia (0.015–0.042 mg GAE/g) [37], honey in southern Italy (0.120 mg GAE/g) [38], Slovenian honey (0.045–0.232 mg GAE/g) [39], or Polish honey (0.13–0.69 mg GAE/g) [15]; and also as very high, such as in Sicilian black honey 0.165–1.33 mg GAE/g [40]. In general, the average polyphenol content in the coffee honey group (0.642 mg GAE/g) is relatively high. This fact was supported by some previous publications on the effect of altitude on polyphenol contents, in that polyphenol content is positively correlated with altitude [41]. It has been found that a favorable correlation agreement between total polyphenols and antioxidant potential exists in 15 grain samples of Western Himalayan high-altitude buckwheat [41]. The coffee honey was obtained from coffee flowers in Dak Lak grown at an average altitude of 400–800 m above sea level and this may be a factor affecting the polyphenol content. Several previous studies have also proven that when plants are exposed to oxidizing agents, their antioxidant functions can be enhanced to protect plants from free radical damage. Higher altitudes, lower temperatures, and higher UV-B radiation meant that more polyphenols in plants can be accumulated. This may be the reason why the coffee honey samples surveyed had quite high polyphenol contents.

The flavonoid contents in honey samples were investigated with differences between samples. The sample with the highest flavonoid content detected was the CoffeeHC 2 sample (0.037 mg QE/g) and the lowest was in sample Commercial K (0.026 mg QE/g) (Table 2). Similarly to polyphenols, flavonoid content is also affected by the factors such as temperature, precipitation, flora, and soil, etc. Different regions and geographical conditions will result in different flavonoid contents in the coffee honey samples [36]; therefore, there is a statistically significant difference in the flavonoid contents between coffee honey and the commercial coffee honey group, and also a statistically significant difference between coffee honey and other flower honey groups. The average flavonoid content of the coffee honey group (0.034 mg GE/g) was higher than the average content of commercial coffee honey (0.029 mg GE/g) and the other flower honey group (0.029 mg GE/g). The coffee honey group is more prominent than the other groups in terms of flavonoid contents reported in Phacelica honey (0.003–0.014 mg GE/g) [15] and Tulang honey (0.011–0.02 mg GE/g) [37], and this can be considered to be a unique characteristic of coffee honey and contributes to affirming its quality. The high level of total phenolic and flavonoid contents of honey will greatly contribute to the antioxidant activity of these honey samples and alleviate several diseases directly or indirectly related to oxidative stress [5].

### 3.3. Antioxidant Activity

Antioxidant activity is shown by the honeys’ antioxidant activity against DPPH^•^ and ABTS^•+^ radicals. IC_50_ value is a widely used parameter for measuring antioxidant activity. A lower IC_50_ indicates higher antioxidant activity. There were significant differences among the honey samples in terms of their scavenging abilities expressed as IC_50_ of the DPPH and ABTS radical-scavenging activities (Table 2). Commercial H showed the lowest IC_50_ value (17.031 ± 0.27 mg/mL for DPPH^•^, 213.769 ± 1.614 mg/mL for ABTS^•+^), and Commercial T displayed the highest value for DPPH^•^ (1.134 ± 0.34 mg/mL) and Forest Honey for ABTS^•+^ (90.0196 ± 1.188 mg/mL). Among the coffee honey tested, the IC_50_ of DPPH radical-scavenging activities value ranged from 1.134 to 17.031 mg/mL, while the IC_50_ of ABTS radical-scavenging activities value ranged from 115.381 to 213.769 mg/mL. The present analysis had characterized coffee honey as having high contents of total sugar, reducing sugar, TPC and TFC compared to the previous studies. These demonstrate this honey’s ability to scavenge free radicals and its antioxidant activity to create many health benefits for human beings. The results of this study are consistent with the observations made in previously reported studies. These antioxidant activities are reported largely due to the main responsible phenolic components and flavonoids [5,37,42]. Based on the previous studies, honey is known for its contribution to the prevention of a number of acute disorders such as inflammation, cardiovascular disease, diabetes, cancer, helping to protect the liver, pancreas, and eyes [5,43]. These benefits of honey are resulted from their ability to improve the oxidation of tissues, organs, body fluids [44] along with the benefits of low glycemic index sugars in honey. Data on the antioxidant activity also evidence that honey is a rich source of antioxidants [44].

### 3.4. Phenolic Acids Analysis

The contents of phenolic acids were analyzed by HPLC (Figure 1) and the analytical parameters were performed according to the published methods with a few modifications. A fairly good performance was obtained in the prediction of the concentrations of the calibration set data, with the linear coefficient of determination (*R*^2^) varying from 0.9989 to 0.9998 for all the substances, as shown in Table 3.

Most of the 13 honey samples surveyed had all seven phenolic acid components. Gallic acid appeared in high concentrations in all studied honey samples, ranging from 0.037–1.015 mg/kg (Table 4). Among these, the Commercial T sample has the highest content of gallic acid, which is consistent with the results of physicochemical properties and total polyphenols detected (1.015 mg GAE/kg). In addition, the ferulic acid content was also found to be high, in the range of 0.023–0.297 mg/kg. The coffee honey group in Dak Lak had a similarity in ferulic acid content, ranging from 0.193 to 0.276 mg/kg. The Commercial K sample did not have cinnamic acid and the Commercial H sample did not have ferulic acid. The remaining acids were low in contents and did not show predominance. Complete quantitative analytical results are presented in Table 4. In general, the coffee honey samples have all seven quantified phenolic acids, in which the compositions of gallic acid and ferulic acid are quite high and similar. Other phenolic components such as caffeic acid, cinnamic acid, and 2,4-dihydroxybenzoic acid can be considered as non-specific components.

### 3.5. Trigonelline and Caffeine Analysis

Trigonelline and caffeine were detected in all coffee flower honey samples and three other floral honeys were analyzed (Table 5). The content of trigonelline and caffeine in coffee honey samples ranged from 0.314–2.399 mg/kg and 8.946–37.977 mg/kg. Among the analyzed coffee flower honey samples, Coffee HC3 had the highest trigonelline (37.977 mg/kg) and CoffeeHB 2 had the highest caffeine content (2.399 mg/kg). In the group of honeys other than coffee flower honey, the forest honey with the highest trigonelline and caffeine content among the analyzed honeys was 4.541 ± 0.032 and 90.258 ± 0.07, respectively. According to previous scientific reports, the content of trigonelline and caffeine in honey in Vietnam is much lower than that of Colombian coffee flower honey (55.4–85.7 mg/kg, 52–97.7 mg/kg) [36]. When compared with Brazilian coffee honey (12.02 ± 0.81 mg/kg) [8], the caffeine content of Vietnamese coffee honey is higher.

### 3.6. PCA Analysis

According to the results of phenolic composition (Figure 2), honey samples can be divided into two groups. Group 1 is the coffee flower honey samples including two commercial and four coffee honey samples from Dak Lak, and they have similar compositions and contents of caffeic acid, chlorogenic acid, 2,4-dihydroxybenzoic acid, cinnamic acid, gallic acid, and ferulic acid comparable with the Longan honey sample. The average contents of gallic acid and ferulic acid are high, while the phenolic components are quite low, and there were not any specific acids observed in these samples.

The other group consisted of samples with a distinct predominance of one or more quantified phenolic acids. Among these, Commercial T and CoffeeHC 3 were similar in gallic acid, chlorogenic acid, and ferulic acid composition, and commercial sample T stood out with the highest quantity of gallic acid. In this group, the *Tithonia diversifolia* honey and CoffeeHB 1 honey samples were significantly different from the other honey samples. These two samples have similar coumaric acid contents. The unique feature of *Tithonia diversifolia* honey is its presence of 2,4-dihydroxybenzoic acid.

The PCA results for all physicochemical parameters with the content of seven phenolics and two alkaloids in this study were shown in Figure 3, so that the honey samples can be grouped into two different groups with the value of 50.00%. Based on the similarity of physicochemical parameters and the quantified contents of seven phenolic acids, two alkaloid all-honey samples could be divided into two groups. Group 1 included most of the coffee honey, along with the forest honey and *Tithonia diversifolia* honey. The coffee honey samples in group 1 showed that the factors contributing to the characteristics of coffee honey were high contents of phenolic acids, such as gallic acid, chlorogenic acid, and ferulic acid. In addition, the average caffeine content in coffee honey samples is also a prominent feature of this honey. Meanwhile, the remaining phenolic acid components quantified in the study are not specific for coffee honey. The total polyphenol and total flavonoid contents of the coffee honey studied in this study were high, indicating the diverse occurrences of other phenolic compounds. The high phenolic and flavonoid content of coffee honey as a characteristic shows its high health value to humans. When compared with honey from other countries, such as Cuban honey (0.213–0.595 GAEmg/g) [45] or Slovenian honey (0.0448–0.2414 GAEmg/g) [46], the relatively high phenolic content in coffee honey will bring much greater health benefits. The content of phenolics and flavonoids of Vietnamese coffee honey is also higher than that of Brazil coffee honey [8] or the phacelia honeys of Poland [15], *Rhododendron* honeys [47], which indicates that this honey is more valuable than other types of honey or coffee honey from other countries. The physicochemical analysis also showed that coffee honey can be a source of relatively high levels of total and reducing sugars. Meanwhile, group 2 included Commercial K, Commercial H, Longan Honey, and CoffeeHC 1, in which physicochemical indexes and the content of seven phenolic acids were usually low and had no outstanding criteria or components.

## 4. Conclusions

This study determined the physicochemical properties, total phenolic contents, and total flavonoid contents, as well as quantified seven individual phenolic acids presented in coffee honey from Vietnam and some other types of flower honey. It was shown that Vietnamese coffee honey samples are characterized by higher gallic acid, ferulic acid and caffeine contents, as well as high levels of phenolic and flavonoids contents. The presences of all seven phenolic acids and the contents of total and reducing sugars are also interesting factors. The results obtained show the high quality of Vietnamese coffee honey. These specific characteristics of Vietnamese coffee honey mean that they could be a potential energy source because of their relatively high content of total and reducing sugars, along with a good supply of nutrients. The high antioxidant activity confers benefits to human health. The data in this article highlight the relevance of Vietnamese coffee honey as a healthy substance.

## Figures and Tables

**Figure 1 foods-11-00388-f001:**
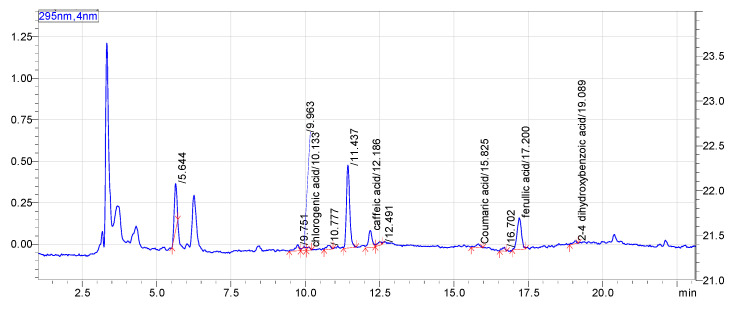
HPLC chromatogram of phenolic acids in coffee honey.

**Figure 2 foods-11-00388-f002:**
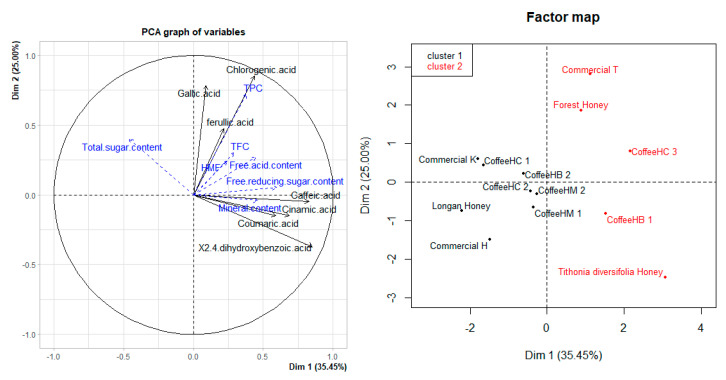
PCA analysis results of seven phenolic acid components of honey samples.

**Figure 3 foods-11-00388-f003:**
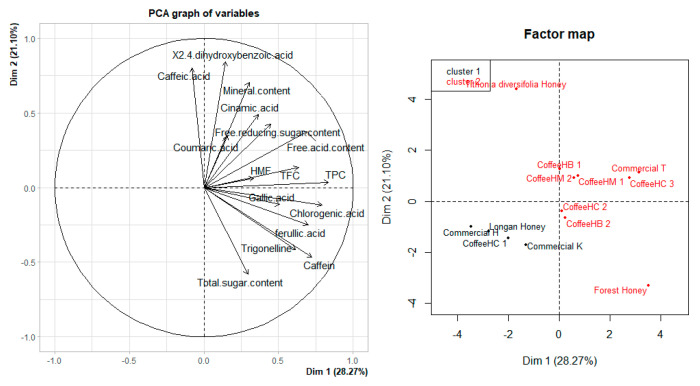
Results of PCA analysis of physicochemical parameters, content of seven phenolic acids and trigonelline, caffein of honey samples.

**Table 1 foods-11-00388-t001:** Results of physicochemical parameters of honey sample.

Name Sample	Physicochemical Parameters				
	Free Acid Content (mg/kg)	Mineral Content (%)	HMF (mg/kg)	Total Sugar Content (mg/kg)	Free Reducing Sugar Content (mg/kg)
CoffeeHC 1	20.326 ± 0.105 ^b^	0.051 ± 0.013 ^ac^	0.048 ± 0.016 ^a^	811.660 ± 2.746 ^bc^	669.964 ± 1.335 ^bc^
CoffeeHC 2	23.902 ± 0.055 ^ef^	0.089 ± 0.044 ^bce^	0.245 ± 0.003 ^b^	884.718 ± 1.762 ^e^	712.129 ± 6.301 ^d^
CoffeeHC 3	29.485 ± 0.259 ^h^	0.131 ± 0.032 ^e^	2.933 ± 0.022 ^h^	804.128 ± 9.591 ^ab^	722.881 ± 5.844 ^de^
CoffeeHB 1	25.692 ± 0.122 ^g^	0.064 ± 0.003 ^acd^	0.108 ± 0.056 ^a^	814.759 ± 3.719 ^bc^	741.375 ± 17.496 ^ef^
CoffeeHB 2	24.327 ± 0.031 ^f^	0.097 ± 0.026 ^ce^	2.722 ± 0.017 ^fg^	848.308 ± 1.093 ^be^	655.251 ± 4.348 ^b^
CoffeeHM 1	31.163 ± 0.779 ^i^	0.095 ± 0.003 ^bce^	2.113 ± 0.047 ^d^	879.702 ± 7.681 ^de^	757.769 ± 1.339 ^f^
CoffeeHM 2	29.455 ± 0.564 ^h^	0.101 ± 0.003 ^ce^	2.037 ± 0.017 ^d^	818.946 ± 15.456 ^bcd^	685.185 ± 2.500 ^c^
Commercial H	16.142 ± 0.296 ^a^	0.041 ± 0.011 ^ab^	2.607 ± 0.036 ^f^	795.115 ± 0.994 ^ab^	625.62 ± 15.746 ^a^
Commercial K	23.185 ± 0.254 ^de^	0.054 ± 0.005 ^acd^	2.712 ± 0.022 ^fg^	828.385 ± 56.895 ^be^	666.972 ± 5.335 ^bc^
Commercial T	26.140 ± 0.179 ^g^	0.107 ± 0.003 ^de^	2.781 ± 0.037 ^g^	831.394 ± 18.746 ^be^	741.878 ± 5.685 ^ef^
Forest Honey	22.318 ± 0.172 ^cd^	0.026 ± 0.007 ^a^	2.624 ± 0.008 ^f^	870.097 ± 12.347 ^ce^	745.118 ± 2.3 ^ef^
*Tithonia diversifolia* Honey	21.647 ± 0.526 ^c^	0.106 ± 0.013 ^de^	2.449 ± 0.038 ^e^	742.860 ± 14.554 ^a^	760.964 ± 1.375 ^f^
Longan Honey	15.220 ± 0.146 ^a^	0.079 ± 0.005 ^ace^	1.123 ± 0.091 ^c^	868.913 ± 37.668 ^ce^	741.414 ± 2.295 ^ef^

The values are mean values of four replicate samples ± standard error of mean and the values with different alphabetical letters represents values significantly different at the 0.05 level of probability according to the ANOVA.

**Table 2 foods-11-00388-t002:** Total polyphenol and flavonoid contents and antioxidant properties of honey samples.

Name Sample	Total Polyphenol (mg GAE/g)	Total Flavonoid (mg QE/g)	IC_50_ of DPPH Inhibition (mg/mL)	IC_50_ of ABTS Inhibition (mg/mL)
CoffeeHC 1	0.519 ± 0.0083 ^a^	0.032 ± 0.0005 ^d^	9.769 ± 0.83 ^f^	175.519 ± 1.440 ^i^
CoffeeHC 2	0.596 ± 0.0048 ^b^	0.037 ± 0.0021 ^f^	10.235 ± 0.86	175.776 ± 1.665 ^i^
CoffeeHC 3	0.704 ± 0.016 ^d^	0.037 ± 0.001 ^f^	4.753 ± 0.60 ^d^	142.955 ± 2.835 ^f^
CoffeeHB 1	0.730 ± 0.025 ^de^	0.030 ± 0.0005 ^c^	3.587 ± 0.89 ^cd^	130.424 ± 1.570 ^e^
CoffeeHB 2	0.657 ± 0.014 ^c^	0.033 ± 0.0007 ^d^	9.669 ± 0.73 ^f^	167.471 ± 2.001 ^h^
CoffeeHM 1	0.749 ± 0.003 ^e^	0.036 ± 0.0003 ^ef^	2.831 ± 0.44 ^bcd^	128.616 ± 0.668 ^e^
CoffeeHM 2	0.863 ± 0.012 ^f^	0.034 ± 0.0003 ^de^	1.589 ± 0.38 ^abc^	123.013 ± 0.721 ^d^
Commercial H	0.508 ± 0.010 ^a^	0.028 ± 0.0003 ^ac^	17.031 ± 0.27 ^g^	213.769 ± 1.614 ^k^
Commercial K	0.699 ± 0.008 ^cd^	0.026 ± 0.0005 ^a^	10.135 ± 0.516 ^f^	158.65 ± 0.286 ^g^
Commercial T	1.090 ± 0.034 ^h^	0.033 ± 0.0005 ^d^	1.134 ± 0.34 ^ab^	115.381 ± 1.665 ^c^
Forest Honey	0.913 ± 0.0053 ^g^	0.032 ± 0.0003 ^d^	1.255 ± 0.236 ^ab^	90.0196 ± 1.188 ^b^
*Tithonia diversifolia* Honey	0.570 ± 0.0025 ^b^	0.029 ± 0.0003 ^bc^	11.855 ± 0.88 ^f^	159.11 ± 1.363 ^g^
Longan Honey	0.512 ± 0.0026 ^a^	0.027 ± 0.0003 ^ab^	15.645 ± 0.572 ^g^	158.535 ± 0.881 ^g^
Ascorbic acid	-	-	0.0000234667 ± 0.00005 ^a^	0.0000164333 ± 0.0000083 ^a^

The values are mean values of four replicate samples (*n* = 3) ± standard error of mean and the values with different alphabetical letters represents values significantly different at the 0.05 level of probability according to the ANOVA.

**Table 3 foods-11-00388-t003:** Validation parameters include linear range, coefficient of determination (R^2^), limit of detection (LOD) and limit of quantitation (LOQ) for each analyte.

Peak Number	Phenolic Acid	Linear Range (mg/L)	R^2^	LOD (mg/L)	LOQ (mg/L)
1	gallic acid	0.1–5	0.9996	0.0063	0.0191
2	chlorogenic acid	0.1–5	0.9989	0.0066	0.020
3	caffeic acid	0.1–5	0.9997	0.0048	0.0146
4	coumaric acid	0.1–5	0.9998	0.0040	0.0123
5	ferulic acid	0.1–5	0.9990	0.0049	0.0149
6	2,4-dihydroxy-benzoic acid	0.1–5	0.9992	0.0322	0.0977
7	cinnamic acid	0.1–5	0.9994	0.0021	0.0066

**Table 4 foods-11-00388-t004:** Phenolic acid contents in honey samples.

Name Sample	Phenolic Acids Concentration (mg/kg)						
	Gallic Acid	Cinnamic Acid	Chlorogenic Acid	Caffeic Acid	Coumaric Acid	Ferulic Acid	2,4 Dihydroxy-benzoic Acid
CoffeeHC 1	1.067 ± 0.005 ^ac^	0.074 ± 0.001 ^ab^	0.794 ± 0.009 ^b^	0.521 ± 0.007 ^ab^	0.223 ± 0.006 ^ab^	1.927 ± 0.015 ^c^	0.406 ± 0.010 ^a^
CoffeeHC 2	1.213 ± 0.008 ^ac^	0.089 ± 0.002 ^ac^	0.44 ± 0.007 ^ab^	1.082 ± 009 ^ce^	0.278 ± 0.011 ^ab^	2.701 ± 0.013 ^de^	0.68 ± 0.003 ^a^
CoffeeHC 3	1.857 ± 0.02 ^cd^	0.134 ± 0.002 ^bc^	1.285 ± 0.02 ^c^	0.220 ± 0.025 ^f^	0.561 ± 0.005 ^b^	2.766 ± 0.013 ^e^	0.858 ± 0.026 ^ab^
CoffeeHB 1	1.935 ± 0.046 ^cd^	0.158 ± 0.003 ^bc^	0.501 ± 0.019 ^ab^	0.996 ± 0.01 ^cd^	1.11 ± 0.005 ^c^	2.141 ± 0.016 ^cd^	0.79 ± 0.036 ^ab^
CoffeeHB 2	1.645 ± 0.027 ^bc^	0.092 ± 0.004 ^ac^	0.79 ± 0.020 ^b^	0.965 ± 0.02 ^cd^	0.279 ± 0.007 ^ab^	1.866 ± 0.013 ^c^	0.59 ± 0.009 ^a^
CoffeeHM 1	1.333 ± 0.034 ^ac^	0.118 ± 0.001 ^ac^	0.344 ± 0.005 ^ab^	1.436 ± 0.005 ^e^	0.149 ± 0.004 ^a^	2.036 ± 0.011 ^c^	0.638 ± 0.010 ^a^
CoffeeHM 2	1.087 ± 0.021 ^ac^	0.133 ± 0.003 ^bc^	0.525 ± 0.024 ^ab^	1.178 ± 0.009 ^de^	0.142 ± 0.0008 ^a^	2.147 ± 0.016 ^cd^	0.687 ± 0.025 ^a^
Commercial H	0.560 ± 0.012 ^ab^	0.147 ± 0.002 ^bc^	0.329 ± 0.002 ^ab^	0.258 ± 0.001 ^a^	0.144 ± 0.009 ^a^	NA	0.578 ± 0.012 ^a^
Commercial K	3.846 ± 0.029 ^e^	NA	0.63 ± 0.018 ^ab^	0.448 ± 0.0063 ^ab^	0.316 ± 0.01 ^ab^	1.599 ± 0.011 ^bc^	0.58 ± 0.006 ^a^
Commercial T	10.124 ± 0.145 ^f^	0.214 ± 0.006 ^c^	1.584 ± 0.02 ^c^	0.695 ± 0.014 ^bc^	0.221 ± 0.0006 ^ab^	1.133 ± 0.02 ^b^	0.761 ± 0.007 ^ab^
Forest Honey	3.796 ± 0.019 ^e^	0.096 ± 0.008 ^ac^	1.294 ± 0.022 ^c^	0.204 ± 0.036 ^f^	0.297 ± 0.001 ^ab^	2.977 ± 0.014 ^e^	0.615 ± 0.009 ^a^
*Tithonia diversifolia* Honey	0.367 ± 0.014 ^a^	0.119 ± 0.009 ^bc^	0.498 ± 0.002 ^ab^	2.381 ± 0.17 ^f^	0.477 ± 0.05 ^ab^	0.377 ± 0.030 ^a^	1.349 ± 0.005 ^b^
Longan Honey	2.981 ± 0.009 ^de^	0.089 ± 0.002 ^ac^	0.23 ± 0.009 ^a^	0.202 ± 0.004 ^a^	0.13 ± 0.004 ^a^	0.236 ± 0.005 ^a^	0.497 ± 0.014 ^a^

The values are mean values of four replicate samples (*n* = 3) ± standard error of mean and the values with different alphabetical letters represents values significantly different at the 0.05 level of probability according to the ANOVA.

**Table 5 foods-11-00388-t005:** Trigonelline and caffeine contents in honey samples (mg/kg).

Name Sample	Trigonelline	Caffein
CoffeeHC 1	0.314 ± 0.003 ^a^	8.946 ± 0.227 ^d^
CoffeeHC 2	1.244 ± 0.012 ^f^	25.736 ± 0.008 ^j^
CoffeeHC 3	2.327 ± 0.007 ^g^	37.977 ± 0.003 ^l^
CoffeeHB 1	0.579 ± 0.013 ^b^	29.954 ± 0.103 ^k^
CoffeeHB 2	2.399 ± 0.007 ^h^	22.319 ± 0.025 ^i^
CoffeeHM 1	0.353 ± 0.015 ^a^	16.597 ± 0.026 ^e^
CoffeeHM 2	0.678 ± 0.020 ^c^	19.163 ± 0.002 ^g^
Commercial H	0.627 ± 0.017 ^bc^	4.556 ± 0.003 ^c^
Commercial K	0.803 ± 0.008 ^d^	21.94 ± 0.010 ^h^
Commercial T	0.928 ± 0.005 ^e^	17.412 ± 0.028 ^f^
Forest Honey	4.541 ± 0.032 ^i^	90.258 ± 0.07 ^m^
*Tithonia diversifolia* Honey	0.957 ± 0.007 ^e^	0.591 ± 0.004 ^a^
Longan Honey	0.763 ± 0.049 ^d^	0.915 ± 0.015 ^b^

The values are mean values of four replicate samples (*n* = 3) ± standard error of mean and the values with different alphabetical letters represents values significantly different at the 0.05 level of probability according to the ANOVA.

## Data Availability

Not applicable.

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
