# Peer review of "Chemical Composition Analysis and Antioxidant Activity of Coffea robusta Monofloral Honeys from Vietnam"

_foods, 2022, doi:10.3390/foods11030388_

Round 1

Reviewer 1 Report

I suggest switching the order of chapters 2.1 to 2.2. and name it appropriately 2.1. General - this name adds nothing. It would be best to remove it altogether and include the necessary information in the chapters describing the methodologies used in the work in detail

chapter 2.3. - is too general, methodologies should be described in such a way that the potential reader can repeat the experience. One cannot be influenced by the fact that each of the readers has access to frequently paid access to publications whose methodology is quoted

line 132 and 141- there is no concept of room tepearature in analytics. some like warmth and their rooms have a temperature of 26 degrees Celsius, while others like cool conditions and it is 18 degrees Celsius. As you know, temperature is a factor that determines the course of chemical reactions. please enter a specific temperature level

line 132- there is no concept of room tepearature in analytics. some like warmth and their rooms have a temperature of 26 degrees Celsius, while others like cool conditions and it is 18 degrees Celsius. As you know, temperature is a factor that determines the course of chemical reactions. please enter a specific temperature level

line 295-305 and 364-373 and 396-403 and 414-433 - first, the text describing the result of the work should appear, then table 2 and 3 and 5 fig 2 not reversed as it is now

table 4 - please reformat the tables in such a way that the results and their standard deviations are in one line. The current layout makes it very difficult to interpret the results and reduces the readability of the work

Reviewer 2 Report

Chemical composition analysis and antioxidant activity of Coffea robusta monofloral honeys from Vietnam

The work describes the properties of coffee honey. The issue is very interesting and there is little scientific evidence about this type of product. The work is very interesting .

  1. In methodology - what does "General" mean? The HPLC method should be described as one of the methods. This needs to be changed. The purity and origin of the used reagent should be given in a separate section.
  2. The discussion of the results and conclusions should emphasize the uniqueness of this honey compared to other honeys.
  3. Figure 3 is difficult to read. It should be corrected.
  4. In the discussion of the results, one should compare the results obtained in this paper with the results in publications more than ten years old.
  5. The method of citing literature in the census is inconsistent with the requirements of the journal. Please change it.

I believe that the article should be applied with these corrections. I only have doubts if the section in which it is to be published is adequate: Plant Food: Bioactive Compounds from Plant Origin and Therapeutic and Nutraceutical Properties for Human Health? Honey is an animal product.
